# The *SMIM25-COX-2* Axis Modulates the Immunosuppressive Tumor Microenvironment and Predicts Immunotherapy Response in Hepatocellular Carcinoma

**DOI:** 10.3390/cimb47090693

**Published:** 2025-08-27

**Authors:** Zhenxing Wang, Xia Li, Shiyi Zhang, Jiamin Sun, Qinchen Lu, Yuting Tao, Shuang Liang, Xiuwan Lan, Jianhong Zhong, Qiuyan Wang

**Affiliations:** 1Guangxi Key Laboratory for Genomic and Personalized Medicine, Guangxi Medical University, Nanning 530021, China; wangzhenxing@stu.gxmu.edu.cn (Z.W.); lixiadeya@163.com (X.L.); shiyizhang990622@163.com (S.Z.); sjm18172067558@163.com (J.S.); luqinchen@stu.gxmu.edu.cn (Q.L.); tytxzxh@126.com (Y.T.); 202320183@sr.gxmu.edu.cn (S.L.); 2Department of Clinical Laboratory, The First Affiliated Hospital of Guangxi Medical University, Nanning 530021, China; 3Department of Biochemistry and Molecular Biology, School of Basic Medical Sciences, Guangxi Medical University, Nanning 530021, China; lanxiuwan@163.com; 4Department of Hepatobiliary Surgery, Guangxi Medical University Cancer Hospital, Nanning 530021, China

**Keywords:** HCC, SMIM25, immunosuppressive microenvironment, COX-2, immunotherapy

## Abstract

Hepatocellular carcinoma (HCC) is a malignancy that is notorious for its dismal prognosis. Dysregulation of the tumor microenvironment (TME) in HCC has emerged as a key hallmark in determining disease progression and the response to immunotherapy. The aim of this study was to identify novel TME regulators that contribute to therapeutic resistance, thus providing mechanistic insights for targeted interventions. The expression of *SMIM25* was evaluated in the the Cancer Genome Atlas-Liver Hepatocellular Carcinoma(TCGA-LIHC) and Guangxi HCC cohorts, and its clinicopathological significance was assessed. RNA sequencing and bioinformatics analyses were performed to elucidate the potential impact of elevated *SMIM25* levels. Immunohistochemistry (IHC) and single-cell mass cytometry (CyTOF) were employed to examine the cellular composition of the tumor microenvironment. The biological effects of *SMIM25* on cell proliferation and migration were studied in vitro using 3-(4,5-Dimethylthiazol-2-yl)-2,5-diphenyltetrazolium Bromide(MTT) and wound healing assays, while its impact on tumor growth was evaluated in vivo in a nude mouse model. Transcriptomic and single-cell proteomic analyses were integrated to explore the mechanism by which *SMIM25* affects the progression of HCC. The expression of *SMIM25* was significantly up-regulated in both HCC tissues and cell lines (*p* < 0.05). RNA sequencing analyses revealed a significant positive correlation between *SMIM25* expression and immunosuppression, and between *SMIM25* expression and extracellular matrix(ECM)-related molecular features. Single-cell mass cytometry revealed two immunosuppressive cell clusters that were enriched in HCC patients with high *SMIM25* expression. Moreover, *SMIM25* was associated with immune exclusion and ECM remodeling signals in the TME of HCC. *SMIM25* overexpression was associated with the expression of the tumor inflammatory marker *cyclooxygenase-2*(*COX-2*), and a *COX-2* inhibitor could partially reverse the biological phenotype associated with *SMIM25* expression in HCC cells (*p* < 0.05). Further transcriptome analysis in immunotherapy cohorts suggested the *SMIM25-COX-2* axis might have predictive value for the response to immunotherapy. Our results suggest that *SMIM25* may serve as a biomarker for the prognosis of HCC patients and may also be a predictive biomarker for the response to immunotherapy, enabling more precise and personalized HCC treatment.

## 1. Introduction

Hepatocellular carcinoma (HCC) is a highly prevalent malignancy of the digestive system, ranking sixth globally among all cancers by incidence, and accounting for the fourth-highest cancer-related mortality worldwide [1]. The HCC mortality rate has been steadily increasing at an annual rate of 2–3%, and a substantial proportion of patients are diagnosed at advanced stages [2]. Despite notable advances in the clinical management of HCC, patient prognosis remains dismal. This is mainly due to the inherent heterogeneity of the tumor mass and the complex architecture of the tumor microenvironment (TME). The TME interacts dynamically with malignant cells and plays crucial roles in facilitating HCC progression and regulating tumor growth, metastasis, and clinical outcomes [3].

In recent years, cancer immunotherapy has demonstrated significant clinical efficacy in various advanced malignancies. In particular, immunotherapy with immune checkpoint inhibitors has shown encouraging results in certain cancer types, notably melanoma and lung cancer [4,5]. Given its immunogenic nature and the co-existence of an immunosuppressive microenvironment, HCC is an appealing candidate for immunotherapy [6]. Immune checkpoint inhibitors, especially those targeting the *programmed cell death protein 1* (*PD-1*) and its ligand programmed death-ligand 1(*PD-L1)*, have exhibited notable clinical efficacy in HCC, highlighting their potential therapeutic value in this malignancy [7,8]. In HCC patients who previously received sorafenib treatment, the objective response rates of anti-*PD-1* blocking immunotherapies such as camrelizumab, nivolumab, and pembrolizumab have reached 15% to 20%. This represents a substantial, three-fold enhancement compared to sorafenib monotherapy [9]. However, the heterogeneous nature of the TME in HCC has limited the effectiveness of immunotherapy [10]. This heterogeneity, both spatially and between different patients, is manifested not only in the varying clinical responses observed between individual patients, but also in different tumor regions within the same patient. Therefore, there is an urgent need for biomarkers that can predict the therapeutic response of HCC patients to immunotherapy.

Given the limitations of current treatments, the development of novel therapeutic modalities is of paramount importance. Among the most promising are peptide-based therapeutics, which offer unique advantages in cancer treatment due to their high specificity and favorable safety profiles [11]. Specifically, in the field of cancer immunotherapy, a major focus is on developing peptide and small-molecule modulators that target critical immune checkpoint proteins such as *PD-1* and *PD-L1*, thereby unleashing the body’s own anti-tumor immune response [12]. Notably, a subclass of therapeutic peptides known as oncolytic peptides has shown exceptional potential, exhibiting a powerful synergistic effect by directly lysing cancer cells while simultaneously stimulating an anti-tumor immune response. The success of these emerging peptide-based strategies, much like existing immunotherapies, relies heavily on identifying novel molecular targets within the TME [13,14,15].

In this study, we identified a long non-coding RNA (lncRNA) called *SMIM25*, which has not been previously reported in the context of HCC. By integrating transcriptome data with single-cell proteomic analyses, we found significant positive correlations between *SMIM25* expression and both immunosuppressive processes and signaling pathways associated with the extracellular matrix (ECM). We also performed a series of biological experiments to elucidate the oncogenic functions of *SMIM25*. Notably, *SMIM25*-associated phenotypes could be partially reversed through inhibition of *COX-2*. These findings provide novel insights into the immunological mechanisms underlying the progression of HCC. Moreover, they deepen our understanding of therapeutic resistance mechanisms within the TME of HCC.

## 2. Materials and Methods

### 2.1. Patients and Samples

A total of 58 HCC tissue samples were collected from HCC patients who underwent potentially curative hepatic resection at Guangxi Medical University Cancer Hospital, China, from 1 January 2018 to 30 November 2020 (Guangxi cohort 1). A total of 74 HCC tissue samples were collected from HCC patients who underwent potentially curative hepatic resection at Guangxi Medical University Cancer Hospital, China, from January 1 to 30 November 2021 (Guangxi cohort 2). Written informed consent was provided by all participants. The collection and use of these samples were approved by the Ethics Committee of Guangxi Medical University (Approval No.: 20200137). The collection and handling procedures strictly adhered to institutional standards and the principles of the Declaration of Helsinki. Comprehensive clinical and pathological information was gathered on age, sex, tumor number, tumor size, Barcelona Clinical Liver Cancer stage [16,17], Edmondson grade [18,19], microvascular invasion, and liver cirrhosis, as well as levels of CK19, aspartate transaminase, alanine aminotransferase, and alpha fetoprotein. In addition, whole-blood samples of HCC patients (*n* = 40) and healthy volunteers (*n* = 40) were prospectively collected from Guangxi Medical University Cancer Hospital from 1 November to 15 November 2021. The collection and use of these samples were approved by the Ethics Committee of Guangxi Medical University (Approval No.: 20200137). All HCC patients were received curative hepatic resection, diagnosed with HCC by two experienced pathologists and enrolled with informed consent. Patients who received neoadjuvant therapy or palliative hepatic resection were excluded. All experiments were performed according to relevant guidelines and regulations. Healthy controls met stringent criteria: (a) absence of chronic liver disease history with confirmed HBV/HCV sero-negativity; (b) normal hepatic function (ALT < 40 U/L, AST < 35 U/L) and AFP levels (<20 ng/mL); (c) no hepatotoxic medication use within 3 months. Exclusion criteria included: (a) blood transfusion/major surgery within 6 months; (b) documented immune disorders or inherited metabolic diseases. All participants, including healthy controls, provided written informed consent using ethics committee-approved forms, with explicit acknowledgment of biobanking purposes. The collection and handling procedures strictly adhered to institutional standards and the principles of the Declaration of Helsinki.

### 2.2. RNA Extraction and Reverse Transcription–Polymerase Chain Reaction (RT-PCR)

RNA was extracted from tissues, whole blood, and liver cancer cells using TRIzol reagent (Invitrogen, Carlsbad, CA, USA), strictly following the manufacturer’s guidelines. Following this, ribosomal RNA was removed from the total RNA samples with the aid of the Ribo-Zero rRNA Removal Kit (Illumina, CA, USA). The resultant purified RNA was then converted into cDNA using the PrimeScript™ RT Reagent Kit provided by Takara (Dalian, China). To determine the expression levels of SMIM25, RT-PCR was performed using SYBR Green Mix (Roche, Boston, MA, USA) along with the specified primer pairs: for *GAPDH*, the forward primer 5′-GTGAACCATGAGAAGTATGACAAC-3′ and the reverse primer 5′-CCAGGACAAGTGCCCAGTA-3′; for SMIM25, the forward primer 5′-CCAGGACAAGTGCCCAGTA-3′ and the reverse primer 5′-TCGGCTCATAGAGTTGTTCAT-3′. The PCR cycling procedure involved an initial denaturation step at 95 °C for 300 s, followed by 40 cycles consisting of denaturation at 95 °C for 10 s, annealing at 60 °C for 10 s, and extension at 72 °C for 10 s. All experiments were conducted in triplicate, with *SMIM25* expression levels normalized to *GAPDH* and analyzed using the 2^−^^ΔΔCT^ method.

### 2.3. RNA In Situ Hybridization (ISH) Assay

The RNA ISH assay was conducted adhering to the guidelines provided by the Boster kit (MK1030,Boster Biological Technology, Pleasanton, CA, USA). Standard procedures were employed for dewaxing paraffin sections to water, followed by a prehybridization step with prehybridization solution maintained at 42 °C for a duration of 2 h. Subsequently, the sections were incubated with a solution containing digoxigenin-labeled probes at 37 °C for 4 h. Post-incubation, the slides underwent rigorous washing and were then subjected to a streptavidin-peroxidase reaction system. They were stained with DAB for a period of 1 min. Counterstaining was performed using 0.1% hematoxylin sourced from Solarbio (Beijing, China) for a duration ranging from 30 s to 1 min. After permeabilization with xylene, the slides were sealed using neutral gum. After permeabilization with xylene, the slides were sealed using neutral gum. Following staining, slides were scanned using a Nano Zoomer S60 Digital slide scanner (HAMAMATSU, Hamamatsu City, Japan). Quantification was performed using ImageJ software (version 1.54k). Each image was scored based on staining intensity, categorized into three levels: negative, weakly positive, and strongly positive [20]. The expression differences between tumor tissues and adjacent normal tissues were then compared based on these scores.

### 2.4. Immunohistochemistry

Antibodies for anti-*CD4* (1:1000 dilution, abcam, ab133616), anti-*CD8* (1:400 dilution, abcam, ab101500, Cambridge, UK), anti-*CD68* (1:8000 dilution, abcam, ab955, Cambridge, UK), anti-*Collagen_1* (1:2000 dilution, abcam, ab138492, Cambridge, UK), anti-*α-SMA* (1:1000 dilution, abcam, ab5694, Cambridge, UK), and anti-*Collagen_5* (1:2000 dilution, abcam, ab7046, Cambridge, UK) were used. The IHC procedure was conducted in the manner outlined below. Initially, the tissue sections underwent baking, deparaffinization, and subsequent rehydration. These sections were then subjected to antigen retrieval through a high-pressure cooking process in an EDTA-based solution (pH = 9) for approximately 30 min. To neutralize endogenous peroxidase activity, the sections were incubated with a peroxidase inhibitor (SP-9000, ZSGB, Beijing, China) at 37 °C for 15 min. Primary antibody staining was carried out at 37 °C for approximately 1.5 h. Following thorough washing, the tissue sections were incubated with Biotinylated Goat Anti-Mouse/Rabbit IgG Polymer (SP-9000, ZSGB, Beijing, China) at room temperature for 30 min. Subsequently, immunostaining was achieved using diaminobenzidine tetrahydrochloride (ZLI-9018, ZSGB, Beijing, China), followed by counterstaining with hematoxylin. For the purpose of image analysis, the ImageJ software (version 1.54k) was employed. The stained sections were scanned using Nano Zoomer S60 Digital slide scanner (Hamamatsu Photonics, Hamamatsu City, Japan). Quantitative analysis was subsequently performed using ImageJ software to evaluate protein expression levels (the specific details can be found in Section 2.14 Quantitative Analysis Using ImageJ).

### 2.5. Cell Culture

The HCC-LM3 (LM3) and THLE2 cell lines were obtained from the Stem Cell Bank of the Chinese Academy of Sciences (Shanghai, China). All cell lines underwent rigorous mycoplasma testing via PCR-based methods, with results confirming negative status. These cell lines were subsequently cultivated and maintained in Dulbecco’s Modified Eagle Medium (DMEM; Gibco, Guangzhou, China), which was enriched with penicillin (100 U/mL), streptomycin (0.1 mg/mL), and 10% fetal bovine serum (FBS; Gibco, Guangzhou, China). Cells were passaged at 80–90% confluence every 2–3 days at 1:4 ratio through enzymatic digestion using TrypLE solution (Life Technologies, Carlsbad, CA, USA) and cultured in an atmosphere containing 5% CO_2_, maintained at 37 °C, and provided with the necessary humidity.

### 2.6. Cell Transfection and Drug Treatment

During the logarithmic growth phase, HCC cells underwent transfection procedures. Prior to transfection, these cells were plated into 6-well dishes, and their density was adjusted to 70%. Subsequently, they were transfected with lentivirus harboring pGWLV11-new (sourced from Genewiz, Suzhou, China), which carries the gene for *SMIM25*. Forty-eight hours post-infection, stable transfected cell lines were selected by the application of puromycin at a concentration of 2 μg/mL (purchased from Solarbio, Beijing, China). Overexpression was verified by quantitative RT-PCR. Two *COX-2*-specific siRNAs, including si-*COX-2*-1 and si-*COX-2*-2 (Genewiz, Suzhou, China), were used to knock down *COX-2*. For transfection, 1.5 × 10^5^ cells/ well were plated into 6-well plates. After cell confluence reached 60%, the *COX-2* siRNA plasmid was transfected into the cells using Lipofectamine™ RNAiMAX (Invitrogen, CA, USA) according to the manufacturer’s instructions. The transfected cells were harvested 48 to 72 h later. The sequences of si-*COX-2*-1 are 5′-GCUGGGAAGCCUUCUCUAATT-3′ (sense) and 5′-UUAGAGAAGGCUUCCCAGCTT-3′ (anti-sense). The sequences of si-*COX-2*-2 are 5′-CCAUCUUUGGUGAAACCAUTT-3′ (sense) and 5′-AUGGUUUCACCAAAGAUGGCA-3′ (anti-sense). The sequences of siRNA-NC are 5′-UUCUCCGAACGUGUCACGUdTdT-3′ (sense) and 5′-ACGUGACACGUUCGGAGAAdTdT-3′ (anti-sense). Three *COX-2*-specific inhibitors, including celecoxib, rofecoxib and valdecoxib (Solarbio, Beijing, China), were dissolved in dimethyl sulfoxide (DMSO, Solarbio) to obtain a stock solution. And the cells were treated with different concentrations of celecoxib, rofecoxib and valdecoxib (0 µM as control and 25 µM as treatment) for 48 h. The knockdown efficiencies of *COX-2* were determined by quantitative RT-PCR.

### 2.7. 3-(4,5-Dimethylthiazol-2-yl)-2,5-diphenyltetrazolium Bromide (MTT) Assay

The HCC cells were plated in 96-well plates at a cellular density of 1500 cells per well. Subsequently, the cells were exposed to two treatment conditions: 0 µM and 25 µM of celecoxib, respectively. At designated time intervals, 10 μL of MTT solution (5 mg/mL; Sigma, New York, NY, USA) was administered to each well, followed by an additional 4-h incubation period. Following this, 100 μL of DMSO was introduced to each well to dissolve the formed formazan crystals. The absorbance values were then determined at a wavelength of 490 nm utilizing a microplate reader (model: FLUOstar Omega, BMG LABTECH, Ortenberg, Germany). All experimental procedures were conducted in triplicate to ensure accuracy and reproducibility.

### 2.8. Scratch Migration Assay

For the scratch migration assay, cells were plated in six-well dishes at a density of 1 × 10^6^ cells per well until they achieved complete confluence. An artificial wound was then induced using a 200 μL pipette tip. Following incubation in serum-free culture medium for durations of 0, 24, or 48 h, magnified images of the wounded area were acquired, and the degree of wound closure was quantified using ImageJ software. The cellular healing rates were determined by calculating the proportion of the scratch area covered by cells. Each experimental condition was replicated three times to ensure reliability and consistency. Migration Reduction (%) = (Area at 0 h − Area at 48 h)/Area at 0 h × 100%.

### 2.9. Animal Experiments

All animal procedures were performed in strict accordance with the Guidelines for the Care and Use of Laboratory Animals and were approved by the Institutional Animal Care and Use Committee of Guangxi Medical University (Approval No:202204017). A total of ten 5-week-old male BALB/c nude mice were sourced from the Laboratory Animal Center of Guangxi Medical University. The animals were housed in individually ventilated cages under specific-pathogen-free (SPF) conditions with ad libitum access to sterile food and water, and were allowed a one-week acclimatization period before the experiment. Using a computer-generated random number list, mice were randomly assigned to two groups (*n* = 5 per group): a control group injected with wild-type LM3 cells (WT-LM3; 2.5 × 10^6^ cells/mouse) and a treatment group injected with LM3 cells overexpressing *SMIM25* (*SMIM25*-OE-LM3; 2.5 × 10^6^ cells/mouse) subcutaneously into the right flank. The investigators responsible for outcome assessment were blinded to the group assignments. Starting on day 8 post-injection, tumor dimensions (length, L; width, W) were measured every two days with a Vernier caliper, and tumor volume was calculated using the formula V = (W^2^ × L)/2. The pre-defined humane endpoint was a tumor size reaching 10 mm × 10 mm or the presence of significant signs of animal distress, such as ulceration, impaired mobility, or more than 20% body weight loss. No animals met exclusion criteria or were removed from the study. On day 16 post-injection, mice were euthanized by CO_2_ inhalation followed by cervical dislocation, and their tumors were harvested, weighed, and processed for further analysis. Data are presented as mean ± standard deviation (SD), and statistical comparisons were made using Student’s t-test, with effect sizes reported as the mean difference with a 95% confidence interval (CI).

### 2.10. Whole—Transcriptome Sequencing

RNA of appropriate concentration and purity was extracted from fresh tumor tissues and sequenced on an Illumina platform. The quality of the sequencing raw data was assessed using FastQC software (v0.11.9) and raw reads with joints, repeats, and low-sequencing quality were removed using Trimmatic (v0.39) [21]. Clean reads were then compared with the human reference genome using Hisat2 software (v2.2.1) [22]. StringTie (v2.1.4) was used to assemble the transcripts and predict their expression [23].

### 2.11. Differentially Expressed Genes and Enrichment Analysis

The DESeq2 package within the R environment was employed to discern differentially expressed genes(DEGs), characterized by expression change of |log2 (fold change)| ≥ 1.5 with a false discovery rate (FDR)-adjusted *p* < 0.05. To delve into the enrichment of Gene Ontology (GO) annotations and Kyoto Encyclopedia of Genes and Genomes (KEGG) pathways, the online resources DAVID 6.8 (accessible at https://david.ncifcrf.gov/ (accessed on 15 March 2021)) and KOBAS 3.0 (located at http://kobas.cbi.pku.edu.cn/ (accessed on 15 March 2021)) were utilized, respectively. Furthermore, gene set enrichment analysis (GSEA) was conducted using the “c5.go.v7.4.symbols” collection, aiming to identify signaling pathways that were enriched in groups exhibiting either high or low expression levels of *SMIM25*. In this context, enrichment was considered significant when the normalized enrichment score surpassed 1, accompanied by a *p*-value below 0.05.

### 2.12. Single-Sample GSEA (ssGSEA)

Based on the results of RNA sequencing, ssGSEA was performed to quantify 27 immune signatures [24] involving immune cells and immune-related pathways and functions. The enrichment score of each sample was then used to classify the 58 HCC tissue samples into patterns of high or low immune cell infiltration.

### 2.13. Cytometry Based on Time of Flight (CyTOF)

#### 2.13.1. Preparation of Single-Cell Suspensions

Excised tumor tissues were promptly immersed in pre-chilled transfer buffer comprising DMEM, supplemented with 2% human serum and 1% penicillin/streptomycin, and subsequently stored in an iced container. For the preparation of single-cell suspensions, the tumor tissues underwent initial mincing into minute fragments, no larger than 1 mm^3^, using a sterile surgical blade. These fragmented tissues were then transferred into 10 mL of digestion medium, which contained a mixture of collagenase lysate enriched with 20 micrograms per milliliter of DNase I (Sigma, New York, NY, USA) and 2 mg per milliliter of collagenase II (Gibco, Guangzhou, China). The suspension was incubated at 37 °C for a duration of 30 min, with manual agitation provided every 5 min to ensure thorough digestion.

Following incubation, the tissue suspension contained within the C tube was dispersed into individual cells through the utilization of the Gentle MACS dissociator (manufactured by Miltenyi, Bergisch Gladbach, Germany), employing a pre-installed protocol (designated as h_tumor_01). The C tube was then inverted and securely attached to the sleeve of the GentleMACS dissociator. Subsequently, the h_tumor_02 program of the GentleMACS was executed once more, and the suspension was further incubated at 37 °C for an additional 30 min. Subsequently, 80% of the cell suspension was aspirated for future applications, whereas the remaining cells underwent processing with the m_imptumor_01 program, utilizing the separator for this purpose.

Subsequently, the tissue suspension was filtered on ice through a 70-micrometer cell strainer sourced from Falcon (Shanghai, China). The cells were then pelleted via centrifugation at 1500 revolutions per minute for a duration of five minutes, after which the supernatant was decanted and discarded. Subsequently, the cells were incubated with three volumes of red blood cell lysis buffer manufactured by Solarbio and transferred into a 2-mL low-bind tube. Following a 10-min incubation period on ice, the cells were centrifuged at 450× *g* for five minutes at a temperature of 4 °C. Upon removal of the supernatant, the cell pellet was resuspended in pre-chilled phosphate-buffered saline. The cell count was subsequently conducted using the trypan blue exclusion method provided by Solarbio, and the cells were cryopreserved at a density of approximately 3 million cells per vial. For the purpose of mass spectrometry staining detection, the cells were frozen by the addition of a mixture comprising 90% fetal bovine serum (FBS) and 10% dimethyl sulfoxide (DMSO). Throughout the dissociation process, the cells were kept on ice whenever feasible to maintain their integrity. The entire procedure was executed in under one hour (typically approximately 45 min) to minimize the risk of experimental errors.

#### 2.13.2. Antibody Labeling and Cell Staining

A panel of thirty-five antibodies (Appendix A) was employed to evaluate the expression profiles of immune cell markers, which were subsequently conjugated to metal-tagged polymers utilizing a 20-well barcoding strategy that incorporated distinctive permutations of six barcoding metals: 102Pd, 104Pd, 105Pd, 106Pd, 108Pd, and 110Pd (sourced from Fluidigm, CA, USA). To distinguish viable from non-viable cells, the specimens were labeled with 5 mmol/L cisplatin (Fluidigm), fixed with 1.6% paraformaldehyde at room temperature for a duration of 10 min, stained with a DNA intercalator (Fluidigm), and then incubated at 4 °C overnight. Following thorough washing with Maxpar Cell Staining Buffer (Fluidigm) and milli-Q water (Merck, Darmstadt, Germany), all samples were resuspended in 0.1 × EQ beads and passed through a 35-micrometer nylon mesh for further processing.

#### 2.13.3. CyTOF Data Acquisition and Analysis

Cell samples were detected using a Helios 2 CyTOF mass spectrometer/flow cytometer with a detection speed < 500 cells/s. The collected flow cytometry data were then standardized and processed with CyTOF software v6.7, and uploaded onto a cloud-based platform (Cytobank (www.cytobank.org (accessed on 11 may 2021))) for in-depth data mining and analysis. The uploaded data were also assessed for their quality, and the antibody names and metal labels were defined. The t-distributed stochastic neighbor embedding map and other algorithms were used for dimensionality reduction and visual data analysis. Associations between certain immune cell subsets and prognosis were identified based on patients’ clinicopathological data.

#### 2.13.4. Imaging Mass Cytometry(IMC)

Liver tissue specimens were sourced from the Pathology Department at Guangxi Medical University Cancer Hospital. The antibody (Appendix A) application and metal conjugation procedures mirrored those utilized in cytometry by CyTOF analysis. Subsequent to dewaxing with xylene, the tissue sections underwent a serial rehydration process involving graded aqueous ethanol solutions (100%, 95%, 80%, and 70%) and were then subjected to antigen retrieval at 96 °C for 30 min. Following cooling to room temperature, the sections were rinsed with double-distilled H_2_O and phosphate-buffered saline for 10 min each, and blocked with 3% bovine serum albumin in PBS at room temperature for an hour. After removal of the blocking solution, the antibody was added dropwise onto the tissue sections. Following overnight incubation at 4 °C, the sections were washed four times with PBS and then labeled with Intercalator-Ir (Fluidigm) diluted 1:400 in PBS at room temperature for 30 min. After another rinse with double-distilled H_2_O, the sections were air-dried for a minimum of 30 min to prepare for mass cytometry imaging.

The resulting data were exported in MCD file format and visualized through the MCD viewer software provided by Fluidigm. To enhance the distinction between antibody signals and background noise, each marker was visually inspected, and a threshold for minimum signal intensity was established within the Fluidigm MCD viewer. The sample image was cropped to facilitate independent analysis of distinct scanning regions, with each original image being imported into histoCAT (version 1.75), a software tool designed for multichannel tissue imaging data analysis. To delineate cell subpopulations exhibiting distinct phenotypic traits, cells were clustered based on their protein expression profiles utilizing the PhenoGraph algorithm.

### 2.14. Quantitative Analysis Using ImageJ

For immunohistochemically stained sections, 2–4 high-resolution images per patient (encompassing tumor and stromal regions) were analyzed using the IHC Profiler plugin integrated in ImageJ. The plugin automatically quantified optical density (OD) values through a standardized workflow: (1) color deconvolution, (2) intensity scoring, and (3) calculation of positive area percentage.

For imaging mass cytometry data, tumor and stromal regions were differentiated based on epithelial cell (Arginase) and type I collagen staining patterns. The workflow comprised: Threshold-based segmentation: Raw images were processed via Image > Adjust > Threshold, with manual adjustment of sliders to isolate target regions, followed by Apply to generate binary masks. ROI management: Selections were converted to regions of interest (ROIs) using Edit > Selection > Create Selection and stored in the ROI Manager (Analyze > Tools > ROI Manager). Quantification: Target fluorescence channels were activated in the ROI Manager window. After toggling Show All to visualize ROIs, integrated density (IntDen) was measured via Measure.

### 2.15. CMap Connectivity Analysis for Drug Prediction

Differentially expressed genes (DEGs) between *SMIM25* high-expression and low-expression groups were identified using DESeq2 (version 1.38.3) with thresholds of log2 fold change (log2FC) ≥ 1 and *p*-value < 0.05. The top 150 upregulated and 150 downregulated genes were selected to construct the disease signature matrix (dis_sig.csv). The refined gene list was submitted to the CMap web portal (https://clue.io/ (accessed on 11 may 2025)). Human hepatocellular carcinoma cell lines were selected for liver disease relevance. The top 40 compounds showing inverse correlation with *SMIM25*-associated signatures were retained (Appendix A).

### 2.16. Statistical Analysisy

Data were expressed as mean ± standard deviation (SD). Statistical analyses were conducted utilizing SPSS version 25.0 (Chicago, IL, USA) and GraphPad Prism version 8.0 (GraphPad Software, Chicago, IL, USA). Intergroup differences were assessed using the independent *t*-test. The chi-squared test was applied to assess associations between categorical clinicopathological characteristics and *SMIM25* expression levels. The DESeq2 package in R software (version 3.6.1) was leveraged to pinpoint differentially expressed lncRNAs, which were defined as those exhibiting an expression change of |log2 (fold change)| ≥ 1.5 with a false discovery rate (FDR)-adjusted *p* < 0.05. A priori power analysis was performed using G*Power 3.1.9.7 to determine the minimal sample size required for detecting clinically meaningful differences. The parameters were defined as follows: two-tailed independent *t*-test, effect size (Cohen’s d) = 0.8, α-error probability = 0.05, and target power = 0.95. This analysis yielded a required total sample size of 70 (35 cases per group), with an achieved power of 0.952. Furthermore, potential links between these lncRNAs and survival outcomes were explored. Statistical significance was attributed to differences with *p* values less than 0.05 (* *p* < 0.05; ** *p* < 0.01; *** *p* < 0.001, **** *p* < 0.0001).

## 3. Results

### 3.1. *SMIM25* as a Potential Prognostic Biomarker for HCC

To identify potential lncRNA biomarkers for the prognosis of HCC, we conducted a comprehensive analysis of differentially expressed lncRNAs in a cohort of 58 pairs of tumor and normal liver tissues utilizing whole-transcriptome sequencing. This revealed a significant upregulation of the lncRNA *SMIM25* in tumor tissues compared to normal tissues (*p* < 0.05; Figure 1A,B). Additionally, in situ hybridization (ISH) experiments confirmed that *SMIM25* expression was significantly higher in tumor tissues compared to adjacent normal tissues and had a predominantly nuclear localization (Figure 1D,E). Consistent with these findings, the expression of *SMIM25* in human hepatoma LM3 cells was observed to be higher than in normal human THLE2 hepatocytes (*p* < 0.05; Figure 1C). Kaplan–Meier analysis revealed that HCC patients in the high-*SMIM25*-expression group had significantly poorer relapse-free survival (RFS; HR = 3.37; 95% CI: 1.12−9.36) and overall survival (OS; HR = 2.13; 95% CI: 1.02−4.43) compared to those in the low-expression group (Figure 1F). Moreover, analysis of the TCGA dataset comprising 341 HCC patients revealed that high expression of *SMIM25* was associated with significantly worse prognosis (HR = 1.51; 95% CI: 1.07−2.13; Appendix A). Subsequently, Chi-squared tests were used to assess the associations between *SMIM25* expression and various clinicopathological characteristics in HCC patients. As shown in Figure 1G,H, high *SMIM25* expression was significantly associated with several high-risk clinicopathological features, including multinodular HCC, advanced tumor stage, higher Edmondson grade, CK19 positivity, and microvascular invasion. Univariate analysis revealed that high *SMIM25* expression was also significantly associated with worse OS (HR = 3.370; 95% CI: 1.213−9.364; *p* = 0.02) (Appendix A). However, Multivariate Cox regression evaluating *SMIM25*’s independent prognostic value (adjusted for *CK19* positivity and MVI status) revealed no significant association with survival outcomes (HR = 2.294; 95% CI: 0.788−6.680; *p* = 0.128), whereas *CK19* positivity remained an independent predictor (HR = 2.758; 95% CI: 1.107−6.876; *p* = 0.029) (Appendix A). Based on our findings at the tissue and cellular levels, we conducted an in-depth analysis of fresh whole-blood specimens sourced from HCC patients and healthy volunteers. This revealed a marked elevation in *SMIM25* expression among HCC patients compared to the control group (*p* < 0.05; Appendix A). *SMIM25* expression exhibited robust discriminatory power, with an AUC value of 0.867 (95% CI: 0.773−0.933) for discriminating between HCC patients and healthy controls. This was associated with a sensitivity of 82.5% and a specificity of 77.5% (Appendix A), indicating that *SMIM25* may serve as a prognosis biomarker for HCC.

### 3.2. Transcriptome Analysis of *SMIM25* Expression and Immunosuppression Markers in HCC

To gain deeper insight into the biological relevance of *SMIM25* in HCC, RNA sequencing was performed to explore functional pathways linked with its upregulation. We stratified 58 HCC patients into *SMIM25*-high (*n* = 29) and *SMIM25*-low (*n* = 29) groups based on the median expression level and conducted differential gene expression analysis between the two groups. A volcano plot identified 2059 DEGs using the criteria of |log2 (fold-change)| ≥ 1.5 and FDR < 0.05), including 1529 upregulated and 530 downregulated genes (Figure 2A). Enrichment analyses of these DEGs in GO terms, KEGG pathways, and GSEA found significant correlations between *SMIM25* expression and signaling pathways related to immune responses and cell proliferation (Figure 2B–D and Appendix A). We also evaluated the correlation between *SMIM25* expression and pathways associated with apoptosis inhibition (e.g., *BCL2A1*, *BIRC5*) and markers of Epithelial–Mesenchymal Transition(EMT) (e.g., *SNAI1*, *VIM*). The results revealed significant upregulation of these pro-tumorigenic genes in the *SMIM25*-high group (*p* < 0.05), thus providing transcription-level evidence that supports a role for *SMIM25* in promoting HCC progression (Appendix A). Correlation analysis identified strong positive associations between *SMIM25* expression and established proliferation/immunosuppression markers (Figure 2E). In addition, HCC patients with elevated *SMIM25* expression demonstrated significantly higher expression levels of immunosuppression-related genes (e.g., *PDCD1*, *CTLA4*, *CD209*) and cell proliferation-associated genes (e.g., *CDK1*, *TTK*, *BUB1B*) compared to patients with low *SMIM25* expression (*p* < 0.05; Figure 2F). Transcriptome analysis revealed that elevated *SMIM25* expression in HCC was associated with immunosuppression.

### 3.3. *SMIM25* Plays a Central Role in Remodeling the Tumor Microenvironment in HCC

Transcriptome profiling revealed significant enrichment of immune-related functionalities in the group with high *SMIM25* expression, prompting us to examine the relationship between *SMIM25* expression and the tumor immune microenvironment. ssGSEA was used to quantitatively assess 27 immune signatures in Guangxi HCC cohort 1. The high-*SMIM25* group exhibited significantly elevated immune activity, particularly in immunosuppressive cell populations such as regulatory T cells (Tregs), compared to the low-*SMIM25* group (*p* < 0.05; Figure 3A,B). The ESTIMATE algorithm, which assesses the stromal and immune cell composition of malignant tumors based on expression data, revealed markedly higher stromal and immune scores in the high-*SMIM25*-expression group compared to the low-expression group (*p* < 0.05; Figure 3C). The ECM plays a crucial role in the TME, providing not only structural support to tumor cells but also fostering an immunosuppressive milieu through interactions with immune cells, thereby enabling tumors to evade the immune system [25]. *SMIM25* expression was positively correlated with collagen-related genes (Figure 3D). In addition, activated stromal-related genes, ECM-stiffness-related genes, and ECM-upregulation-related genes showed positive correlations with *SMIM25* expression (Figure 3E). The *LOX* family (Lysyl Oxidase Family) group of enzymes, involved in the cross-linking of collagen and elastin [26], and the *MMP* family (Matrix Metalloproteinase), responsible for ECM degradation and remodeling [27], also demonstrated positive correlations with *SMIM25* expression (Figure 3F and Appendix A). To obtain a deeper understanding of the relationships between *SMIM25* and ECM remodeling and the immune response, we conducted IHC staining of formalin-fixed paraffin-embedded (FFPE) sections from 58 HCC patients to assess the expression and spatial distribution of immune cells and ECM components. The high-*SMIM25* group exhibited extensive collagen deposition in the area surrounding the tumor, whereas less collagen was observed in the low-*SMIM25* group (Figure 4A,C,D). In line with the transcriptome studies described earlier, patients with high *SMIM25* expression had a significantly higher abundance of immune cells compared to patients with low expression (*p* < 0.05; Figure 4B,D). Notably, collagen deposition in the high-*SMIM25* group appeared to influence the distribution of immune cells, with significantly higher *CD4* and *CD8* expression in the stromal region compared to the tumor region (*p* < 0.05; Figure 4E). IMC provided further evidence of a distinct immune phenotype in the high-*SMIM25* group, with significantly higher *CD4* and *CD68* expression in the stromal region compared to the tumor region (*p* < 0.05; Figure 4F,G). These results indicate the high-*SMIM25* group exhibits immune exclusion and ECM remodeling of the TME.

### 3.4. An Immunosuppression-Related Cell Cluster That Has High *SMIM25* Expression Is Recruited to the TME in HCC

To investigate the impact of *SMIM25* expression on the tumor immune microenvironment, we employed mass cytometry via CyTOF for high-dimensional, single-cell phenotypic profiling of tumor-infiltrating immune cells. From the cohort of 58 patients, we screened 34 tumor tissue specimens that met stringent quality standards for the preparation of single-cell suspensions (17 cases per group in the high- and low-*SMIM25*-expression cohorts). Systematic comparative analysis revealed the existence of a distinct immune cell subset in the two groups. Groups were defined by the median *SMIM25* mRNA level, and patient selection ensured a balanced representation of key clinical characteristics while maintaining consistency with transcriptome stratification. We developed a 35-marker panel encompassing markers for a variety of immune cell types and quantified the expression of each marker within individual cells. Initially, we employed unsupervised t-distributed stochastic neighbor embedding (t-SNE) clustering to differentiate and identify discrete cell populations. An exhaustive analysis of high-dimensional, single-cell data was conducted using the FlowSOM [28] and PhenoGraph [29] algorithms. This resulted in the identification of 17 distinct clusters, each distinguished by unique patterns of marker expression (Figure 5A,B). These clusters included three *CD4*+ T-cell subsets (14, 16, 17), three *CD8*+ T-cell subsets (1, 4, 9), three *CD45RA*+ cell subsets (2, 3, 6), one B-cell subset (7), two Treg subsets (13, 15), one macrophage subset (11), one tumor-associated neutrophil subset (10), one monocyte subset (12), and two unknown T-cell subsets (5,8) (Figure 5A,B). Differential analysis of cluster abundance revealed that clusters 13 and 15 were predominantly enriched in the high-*SMIM25* group (*p* < 0.05; Figure 5C–E). Moreover, the analysis showed these two clusters represented immunosuppressive cells. Notably, cluster 13 exhibited high expression of immunosuppressive checkpoints (*CD279* and *TIM_3*), but low expression of interleukin family cytokines (*IL_6*, *IL_10*, *IL_17A*), suggesting a regulatory T cell (Treg)-like phenotype characterized by potent immunosuppressive activity (Appendix A). Further analysis revealed a strong association between *SMIM25* expression and clusters 13 and 15 (Figure 5F). Both clusters showed a trend toward association with poorer clinical outcomes (*p* > 0.05; Figure 5G). Collectively, these findings suggest that elevated *SMIM25* expression may contribute to the establishment of an immunosuppressive TME in HCC.

### 3.5. *SMIM25* and *COX-2* Appear to Have Synergistic Roles in Promoting HCC Development

We next explored the effects of *SMIM25* on the proliferative and migratory capabilities of HCC by establishing stable HCC cell lines expressing high levels of *SMIM25*. *SMIM25* overexpression in LM3 cells was confirmed with RT-qPCR (*p* < 0.05; Figure 6A). MTT assays revealed that cell proliferation in LM3 cells with *SMIM25* overexpression was significantly enhanced compared to that in wild-type cells (*p* < 0.05; Figure 6B). Additionally, scratch wound healing assays demonstrated that LM3 cells with *SMIM25* overexpression displayed a marked increase in their migratory capacity compared to wild-type cells (Appendix A). In line with these in vitro results, xenograft experiments demonstrated that *SMIM25* overexpression significantly accelerated tumor growth in vivo, with the *SMIM25*-overexpressing group showing a faster tumor growth rate and larger tumor volume compared to the control group (95% CI [0.0018, 0.0749]; Cohen’s d = 1.48; *p* < 0.05; Figure 6C and Appendix A). These results suggest that *SMIM25* plays a pivotal role in the progression of HCC by stimulating cell proliferation, migration, and tumor growth, thereby emphasizing its potential as a significant biomarker of malignancy in HCC.

Given the potential link between *SMIM25* and immunosuppression, we next investigated the correlation between the expression levels of *SMIM25* and immune checkpoints proteins. This analysis revealed significant correlations with *COX-2*, *HAVCR2*, *CD86*, and *PDCD1LG2* (Figure 6D). Notably, induction of *SMIM25* overexpression in LM3 cells resulted in a substantial increase in *COX-2* expression, hinting at a possible association between *SMIM25* and *COX-2* (*p* < 0.05; Figure 6E). To further examine this relationship, we conducted a rescue experiment by knocking down *COX-2* expression in *SMIM25*-overexpressing LM3 cells using two *COX-2*-specific siRNAs and three *COX-2* inhibitors (celecoxib, rofecoxib, and valdecoxib). As expected, downregulation of *COX-2* partially reversed the effects of *SMIM25* overexpression (*p* < 0.05; Figure 6F,G). Given the transient nature of *COX-2* siRNA transfection in *SMIM25*-OE-LM3 cells, we selected celecoxib, the most specific *COX-2* inhibitor, for subsequent functional studies. MTT and scratch wound healing assays showed that *SMIM25*-overexpressing LM3 cells had significantly increased proliferation and migration abilities compared to wild-type LM3 cells. Importantly, these effects were partially attenuated by *COX-2* inhibition in both *SMIM25*-overexpressing and wild-type LM3 cells (*p* < 0.05; Figure 6H–J). This implies a potential correlation between *SMIM25* and *COX-2* in HCC cells, and suggests that *COX-2* inhibition may mitigate the tumor-promoting effects of *SMIM25*.

### 3.6. Predictive Value of the Synergistic *SMIM25-COX-2* Axis in Determining Immunotherapy Response in Cancer

Given the role of *SMIM25* in immune suppression, we next investigated its possible translational relevance to cancer immunotherapy. ssGSEA analyses were performed to assess the relationship between *SMIM25* expression levels and tumor immune infiltrates across a range of different cancer types. This revealed positive correlations between *SMIM25* expression and the infiltration of diverse immune cell populations, particularly Tregs, across numerous cancer types, suggesting the involvement of *SMIM25* in shaping the immunosuppressive TME (Appendix A). Due to the absence of publicly available HCC immunotherapy cohorts with transcriptome data at the time of this study, we validated our model using datasets from bladder cancer, melanoma, and renal cell carcinoma. In patients in the IMvigor210 cohort, high *SMIM25* expression was associated with significantly worse clinical outcomes and reduced overall survival following immunotherapy when compared to low expression (HR = 1.42; 95% CI: 1.01−1.98; *p* < 0.05; Figure 7A). Similar trends were observed in renal cell carcinoma patients (HR = 1.34; 95% CI: 1.03−1.74; *p* < 0.05; Figure 7B). In melanoma, a non-significant trend toward poorer prognosis was observed, with further validation warranted due to the lack of statistical significance (*p* > 0.05; Figure 7C). Samples from the IMvigor210 cohort were subsequently categorized into three distinct immune-microenvironment-associated subtypes: immune-deserted, immune-inflamed, and immune-excluded. Notably, patients with immune-excluded tumors and high *SMIM25* expression exhibited a non-significant trend toward adverse clinical outcomes (*p* > 0.05, Figure 7D), suggesting a potential association that warrants further investigation in larger cohorts. Among the 348 patients enrolled in the IMvigor210 cohort, a variety of responses to anti-*PD-L1* receptor inhibitors were noted, ranging from complete response (CR) to partial response (PR), stable disease (SD), and progressive disease (PD). Interestingly, amongst patients with immune-excluded tumors, those with high *SMIM25* expression were more likely to have either SD or PD compared to those in the low-*SMIM25* group (Figure 7E). Additional investigation into the clinical significance of *COX-2* and *SMIM25* showed that elevated *COX-2* expression was linked to poor prognosis in patients receiving immune checkpoint blockade (ICB) therapy (HR = 1.43; 95% CI: 1.08−1.91; *p* < 0.05; Figure 7F). Furthermore, when *COX-2* and *SMIM25* were simultaneously overexpressed, the prognosis was even worse (*p* < 0.05; Figure 7G). Among patients with high *SMIM25* and *COX-2* levels, a higher frequency of SD or PD was observed compared to patients with high *SMIM25* but low *COX-2* levels, although this difference did not reach statistical significance (*p* > 0.05; Figure 7H). To bridge the translational gap between pan-cancer therapeutic predictions and HCC-specific treatment strategies, we performed Connectivity Map (CMap) analysis using the DEGs from the high- and low-*SMIM25*-expression groups. This approach enabled the identification *JAK* inhibitors (BRD-K04546108) and *COX* inhibitors (Isoniazid) as candidate molecules capable of reversing *SMIM25*-driven immunosuppressive signatures (Figure 7I; Appendix A). These findings highlight the potential of *COX-2*-predicted responses to immunotherapy in HCC patients with high *SMIM25* expression, while also offering valuable insights into the development of novel immunotherapeutic strategies for HCC.

## 4. Discussion

Small integral membrane protein 25 (*SMIM25*), also known as long intergenic nonprotein coding RNA 1272 (*LINC01272*), *PELATON*, or GC-related lncRNA1 (*GCRL1*), is classified as a lncRNA [30,31]. As an oncogenic lncRNA, *SMIM25* has been shown to attenuate mutant-*p53*-induced ferroptosis and enhance mutant *p53*-driven proliferation in Glioblastoma Multiforme (GBM) [32]. Further investigations have shown that *SMIM25* promotes metastasis in gastric cancer (GC) by regulating the expression proteins associated with EMT, suggesting it may be a valuable diagnostic biomarker for GC [33]. Consistent with these findings, our results showed that *SMIM25* expression was significantly upregulated in HCC tissues compared to adjacent non-cancerous tissues, and was also associated with poor prognosis in HCC patients. Moreover, *SMIM25* overexpression was demonstrated to promote cell proliferation, migration, and tumorigenicity in HCC.

To further investigate the impact of *SMIM25* expression on immune cell composition, we performed ssGSEA and CyTOF analyses on 34 HCC tumor tissues. Pathway enrichment analysis revealed that HCCs with high *SMIM25* expression exhibited significant enrichment in signaling pathways related to immune responses and cell proliferation. Furthermore, *SMIM25* expression was positively correlated with that of immunosuppressive genes, including *CTLA4*, *PDCD1*, *CD274*, and *PDCD1LG2*, which encode key immune checkpoint proteins [34]. These immune checkpoints bind to their respective ligands, delivering a “STOP” signal to T cells and attenuating T-cell activation [35]. Notably, CyTOF analysis revealed increased Treg infiltration and accumulation within the TME in HCC patients with high *SMIM25* expression, correlating with poor prognosis and lower survival rates. Tregs suppress effector T cell activity through various mechanisms, including the secretion of immunosuppressive cytokines (e.g., *IL-10* and *TGF-β*) and the expression of immune checkpoint proteins (e.g., *CTLA-4* and *PD-1*) [36,37,38], thereby allowing tumor cells to evade immune surveillance.

Immunotherapy has emerged as a crucial adjunct therapy for cancer by inhibiting immunosuppressive mechanisms and enhancing adaptive T-cell-based therapies [39]. However, immune exclusion can limit patient responsiveness and hinder the efficacy of immunotherapies. A key factor contributing to immune exclusion is the presence of physical barriers that prevent direct interaction between T cells and cancer cells, thereby hindering the activation of immune effector gene signatures. Accumulating evidence suggests that tumor collagens contribute to T-cell exclusion by acting as both physical barriers and a source of immunosuppressive molecules [25,40,41,42]. In the present study, patients with high *SMIM25* expression exhibited significant collagen accumulation within their tumors, leading to immune cell infiltration into the ECM and a phenotype of immune exclusion.

*COX-2* has been shown to play a pivotal role in cancer cells by orchestrating the intra-tumoral inflammatory environment and facilitating tumor progression via immune escape mechanisms [43]. The current research found a direct correlation between the expression levels of *COX-2* and *SMIM25*. Notably, the upregulation of *SMIM25* in LM3 cells resulted in a marked increase in *COX-2* expression, indicating a possible regulatory interaction between these two molecules. Furthermore, *COX-2* inhibition with celecoxib attenuated the proliferation and migration of tumor cells overexpressing *SMIM25*. Consistent with our findings, clinical trials have demonstrated that combining anti-*PD-1* immunotherapy with the *COX-2*-specific inhibitor celecoxib enhanced therapeutic efficacy in colorectal cancer patients [44,45]. Numerous studies have demonstrated that inhibiting *COX-2* can substantially improve the immunotherapy response [46,47]. Furthermore, administration of *COX-2* inhibitors in conjunction with anti-programmed death ligand-1 (*α-PD-L1*) therapy can fully suppress metastatic tumor growth, effectively transforming “cold tumors” into “hot tumors” [48,49].

Our findings are subject to several limitations that warrant further investigation. Although we showed that (1) *SMIM25* was associated with immune suppression and ECM remodeling, (2) *SMIM25* overexpression upregulates *COX-2* expression, and (3) *COX-2* inhibitors partially reverse the biological effects, several critical knowledge gaps remain. First, our mechanistic insights, including the regulatory link between *SMIM25* and *COX-2* and the functional rescue by a *COX-2* inhibitor, were primarily validated in vitro using a single cell line. The therapeutic potential of targeting this axis has not yet been confirmed in an in vivo setting, which represents a crucial next step for future studies. Moreover, the precise upstream regulatory mechanisms governing *SMIM25* expression are still not completely understood. These factors limit the generalizability of our findings to other HCC subtypes and underscore the need for further research. Second, the observed correlation between *SMIM25* and ECM dynamics remains preliminary, and there is a lack of causal evidence linking *SMIM25* as a direct driver of ECM reprogramming. Future research should therefore focus on elucidating the molecular mechanisms by which *SMIM25* influences ECM remodeling. This potentially occurs through direct interaction with ECM components or via modulation of the expression of ECM-related genes. Third, while the *SMIM25-COX-2* axis is a promising immunotherapy target, this hypothesis has not been directly tested for HCC due to the lack of dedicated HCC immunotherapy cohorts. Future studies should therefore prioritize the validation of the *SMIM25-COX-2* axis as a predictive biomarker for immunotherapy response in well-defined HCC immunotherapy cohorts. Furthermore, targeting of the *SMIM25-COX-2* axis in combination with existing immunotherapeutic strategies could offer novel avenues for improving treatment outcomes in HCC patients.

## 5. Conclusions

In summary, our results demonstrate that *SMIM25* expression is associated with survival outcomes in HCC patients. We confirmed the expression of *SMIM25* in tissues, blood, and cells, and elucidated its associated tumor phenotype through functional experiments. Our findings indicate that *SMIM25* expression is correlated with an immunosuppressive microenvironment that is capable of remodeling the TME. Importantly, *SMIM25* and *COX-2* appear to have synergistic roles in promoting HCC development. These findings suggest that the *SMIM25-COX-2* axis could serve as a biomarker for prognosis and immunotherapy response in HCC.

## Figures and Tables

**Figure 1 cimb-47-00693-f001:**
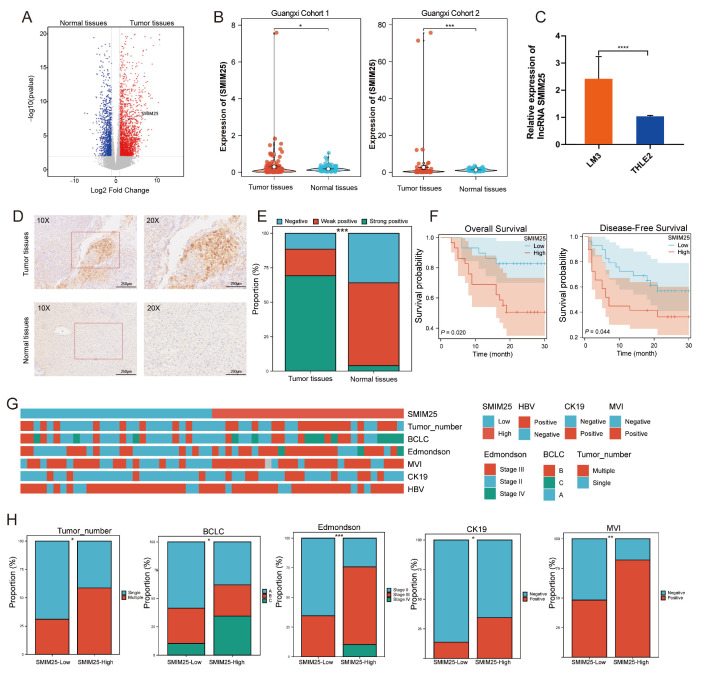
The upregulation of *SMIM25* expression is associated with an unfavorable prognosis in HCC(Hepatocellular carcinoma). (**A**) A volcano plot illustrates the differential expression profiles of lncRNAs between tumor tissues and normal tissues. (**B**) Boxplots compare the expression levels of *SMIM25* in tumor tissues versus normal tissues within a Guangxi HCC cohort. (**C**) *SMIM25* expression in HCC (LM3) versus normal liver epithelial cells (THLE2). (**D**) In situ hybridization (ISH) confirms the distinct expression patterns of *SMIM25* in HCC tissues compared to normal tissues. (**E**) A composite column chart displays the ISH scoring of *SMIM25* in both tumor and normal tissues. (**F**) Analysis of overall survival and disease-free survival underscores the impact of deregulated SMIM25 expression on the prognosis of HCC patients. (**G**,**H**) The heatmap and bar chart demonstrate the comparison of *SMIM25* expression in HCC tissues and clinical–pathological features. * *p* < 0.05; ** *p* < 0.01; *** *p* < 0.001; **** *p* < 0.0001.

**Figure 2 cimb-47-00693-f002:**
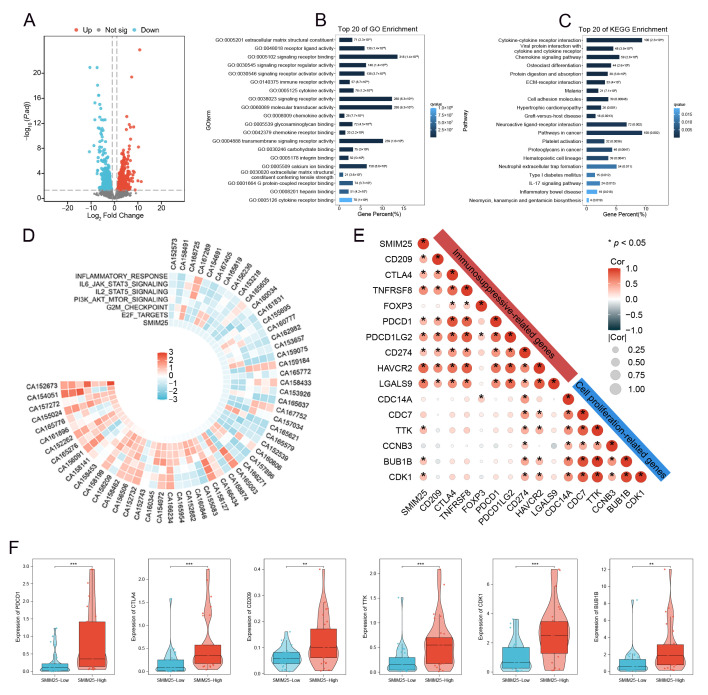
Transcriptome profiling elucidates the link between *SMIM25* expression and immunosuppression in HCC. (**A**) The volcano plot displays DEGs in HCC patients between high and low *SMIM25* expression. (**B**) GO analysis indicates that the top 20 biological pathways are significantly enriched in the high *SMIM25* expression group. (**C**) KEGG analysis reveals that the top 20 biological pathways are significantly enriched in the high *SMIM25* expression group. (**D**) The circular heatmap depicts the variances in the expression of diverse Hallmark pathways between the high and low *SMIM25* expression groups. (**E**) The correlation heatmap displays the correlation between *SMIM25* and genes related to cell proliferation and immunosuppression. (**F**) Boxplots exhibit the expression of cell proliferation and immunosuppressive-related genes in HCC with high and low expression of *SMIM25*. * *p* < 0.05; ** *p* < 0.01; *** *p* < 0.001.

**Figure 3 cimb-47-00693-f003:**
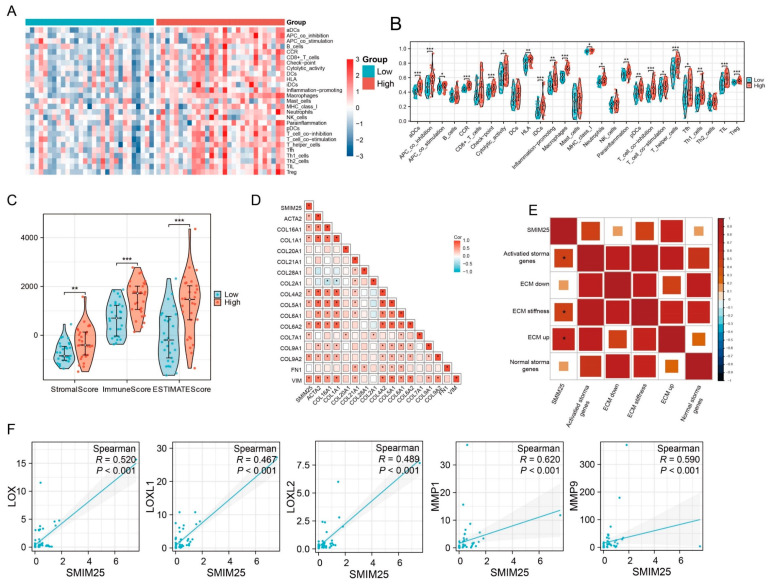
Activation of Extracellular Matrix(ECM) pathways in the high expression of *SMIM25* group. (**A**) A heatmap shows the infiltration patterns of immune cell types between the high and low *SMIM25* expression groups. (**B**) The abundance of immune-associated cells is assessed in both high and low *SMIM25* expression groups. (**C**) A boxplot visualizes the stromal, immune, and ESTIMATE scores between the high and low *SMIM25* expression groups. (**D**,**E**) The correlation heatmap depicts the correlation between *SMIM25* and collagen-related genes, as well as ECM-related signatures. (**F**) Scatter plots reveal the correlation between the expression of *SMIM25* and collagen-modifying enzymes. * *p* < 0.05; ** *p* < 0.01; *** *p* < 0.001.

**Figure 4 cimb-47-00693-f004:**
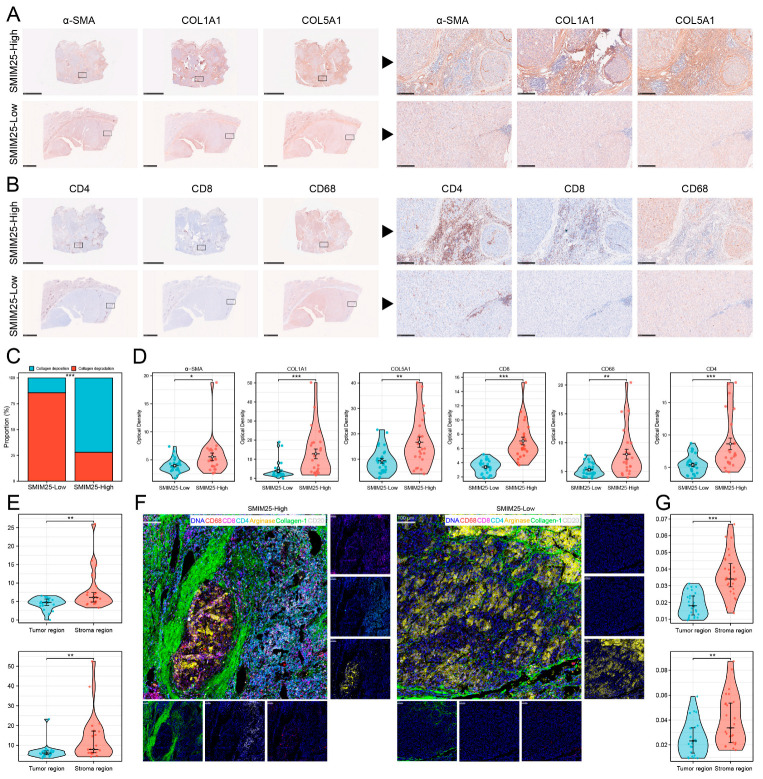
*SMIM25* Expression correlates with immune exclusion and ECM-remodeling in the tumor microenvironment. (**A**–**D**) Immunohistochemical (IHC) analysis is conducted to compare the expression intensity and spatial arrangement of *α-SMA*, *Collagen_1*, *COL5A1*, *CD4*, *CD8*, and *CD68* between the groups exhibiting high and low *SMIM25* expression. (**E**) Boxplots illustrate the variations in *CD4* and *CD8* expression within the stromal and tumor regions of the high *SMIM25* expression group. (**F**) Imaging mass cytometry is utilized to visualize the histological features of HCC with high and low *SMIM25* expression. Representative images showing staining the marker DNA (blue), *collagen_1* (green) and the immune cell marker *CD4* (cyan), *CD8* (purple), *CD68* (red), *PD-L1* (yellow). (**G**) Boxplots compare *CD4* and *CD68* expression levels in stromal versus tumor regions (high *SMIM25* group), quantified by imaging mass cytometry (IMC). * *p* < 0.05; ** *p* < 0.01; *** *p* < 0.001.

**Figure 5 cimb-47-00693-f005:**
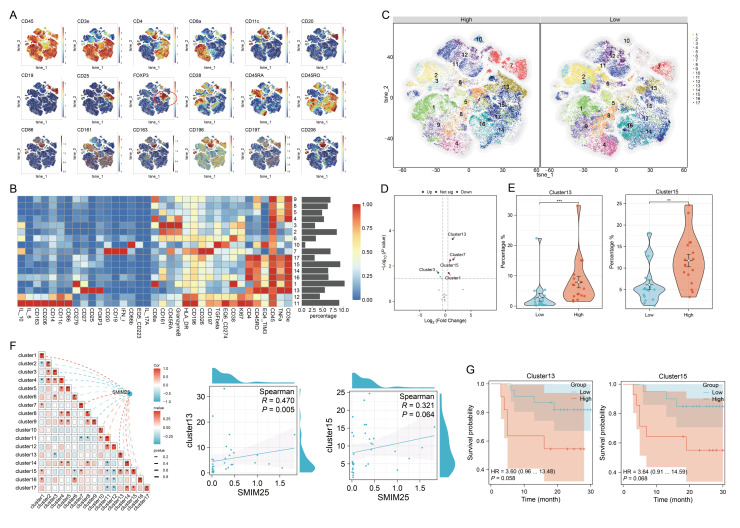
Single-cell Mass Cytometry analysis elucidates the immunosuppressive microenvironment prevalent in the high expression of *SMIM25* group. (**A**) t-SNE maps show the expression profiles of tumor microenvironment-related markers in HCC with high versus low *SMIM25* expression. (**B**) A heatmap depicts the expression patterns of these markers across 17 distinct cell clusters, with bar plots on the right indicating the proportional representation of each cluster within the total cell population. (**C**) t-SNE maps delineate the cellular cluster compositions in HCC compared those with high and low *SMIM25* expression levels,the numbers within the plot denote the different cell clusters. (**D**) Volcano plot portrays the differential abundance of cell clusters in HCC between high and low *SMIM25* expression. (**E**) Boxplots illustrate the proportions of cluster 13 and cluster 15 in HCC between the high and low *SMIM25* expression groups. (**F**) The correlation heatmap and scatter plots exhibit the correlation between the abundance of cell clusters (cluster13 and cluster15) and the expression of *SMIM25*. (**G**) Kaplan–Meier survival curves show the survival outcomes of patients stratified based on the levels of cluster 13 and cluster 15. * *p* < 0.05; ** *p* < 0.01; *** *p* < 0.001.

**Figure 6 cimb-47-00693-f006:**
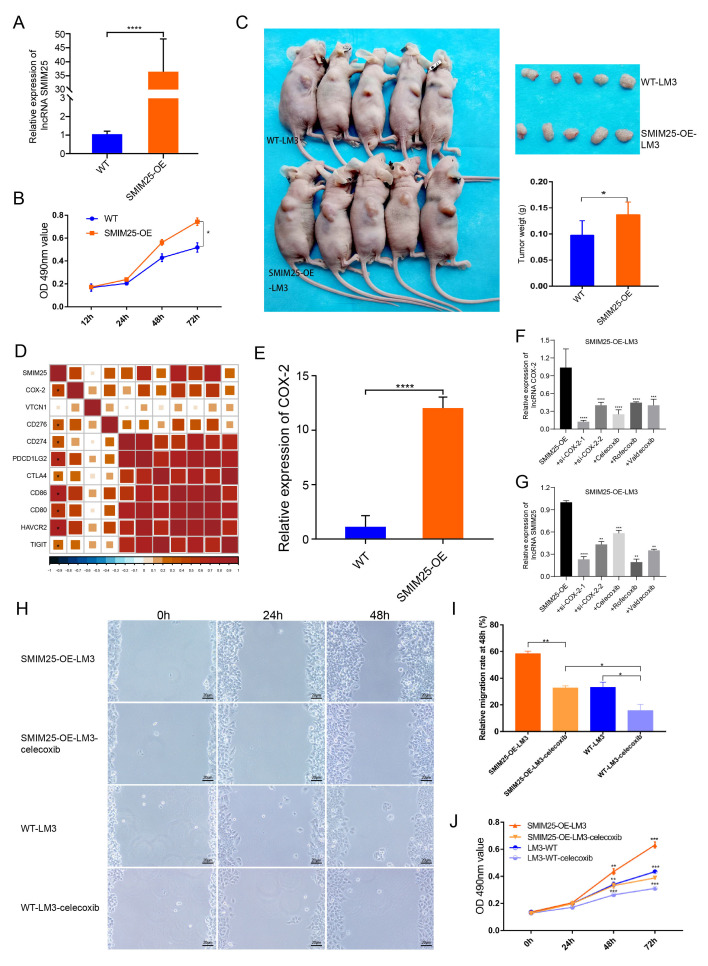
*SMIM25* and *COX-2* may exhibit cooperative effects in accelerating the progression of HCC. (**A**) The efficacy of *SMIM25* overexpression in LM3 cell lines is evaluated using quantitative RT-PCR. (**B**) The proliferative capacity of both wild-type LM3 cells (WT-LM3) and LM3 cells with *SMIM25* overexpression (*SMIM25*-OE-LM3) is assessed through MTT assays. (**C**) Tumor growth and weight are compared in nude mice implanted with either WT-LM3 or *SMIM25*-OE-LM3 cells. (**D**) The correlation heatmap displays the correlation between *SMIM25* and Immune checkpoints. (**E**) The expression levels of *COX-2* are examined in both WT-LM3 and *SMIM25*-OE-LM3 cells. (**F**,**G**) The expression profiles of *COX-2* and *SMIM25* are analyzed when *COX-2* is inhibited. (**H**–**J**) The migration (**H**,**I**) and proliferative (**J**) abilities of both WT-LM3 and *SMIM25*-OE-LM3 cells are evaluated under *COX-2* inhibition, Magnification, 100×. * *p* < 0.05; ** *p* < 0.01; *** *p* < 0.001; **** *p* < 0.0001.

**Figure 7 cimb-47-00693-f007:**
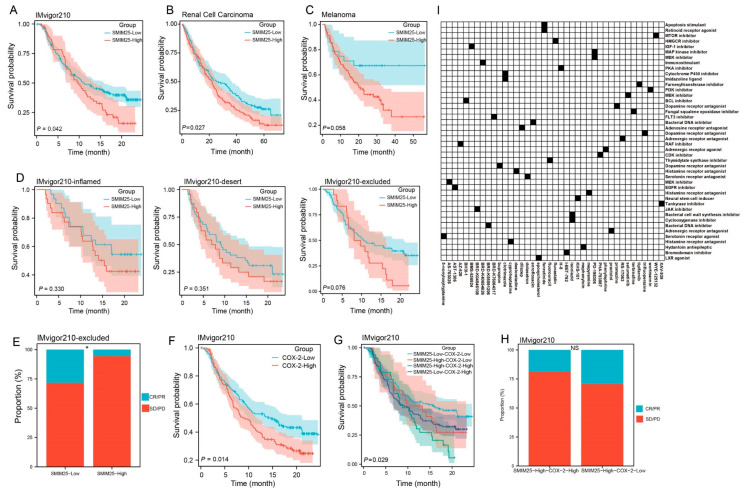
*SMIM25* and *COX-2* as Predictors of Immunotherapy Response. (**A**–**C**) Survival analyses are conducted to compare patients with low versus high *SMIM25* expression in the Bladder cancer (**A**), Melanoma (**B**) and Renal Cell Carcinoma (**C**). (**D**) Kaplan–Meier survival curves depict the overall survival of patients in the IMvigor210 cohort stratified by *SMIM25* expression across three immune phenotypes: inflamed, desert, and excluded. (**E**) Response rate is analyzed in the IMvigor210 cohort (immune-excluded phenotype) based on *SMIM25* expression. (**F**) Survival analysis is performed to compare low and high expression of *COX-2* in Bladder cancer. (**G**) Overall survival curves illustrate the overall survival of for patients stratified by the combined expression levels of *SMIM25* and *COX-2*. (**H**) Response rate is evaluated in the IMvigor210 cohort (immune-excluded phenotype) based on *SMIM25* expression and *COX-2* expression. (**I**) Heatmap displays the top 40 small-molecule compounds predicted by Cmap to exert inhibitory effects on *SMIM25*-High HCC, along with their corresponding mechanisms of action. NS: no significant; * *p* < 0.05.

## Data Availability

All data generated and described in this article are available from the corresponding author on reasonable request.

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
