# Peer review of "The SMIM25-COX-2 Axis Modulates the Immunosuppressive Tumor Microenvironment and Predicts Immunotherapy Response in Hepatocellular Carcinoma"

_cimb, 2025, doi:10.3390/cimb47090693_

Round 1
Reviewer 1 Report
Comments and Suggestions for Authors
Recommend acceptance with minor revisions
Immunotherapy using immune checkpoint inhibitors have been shown to show significant clinical efficacy in HCC, but biomarkers that can predict the treatment response of HCC patients to immunotherapy are needed to avoid heterogeneity of TME. In this paper, the biological effects of SMIM25 on cell proliferation, migration, invasion, and tumor growth were studied by RNA sequencing, IHC, CyTOF, as well as in vivo and in vitro experiments. Especially, SMIM 25 and COX-2 appear to have a synergistic effect in promoting HCC development, suggesting that the SMIM 25-COX-2 axis can be used as a biomarker for HCC prognosis and immunotherapy response.
This manuscript provides a theoretical reference for studying the immunological mechanism of HCC progression, which is of great significance and should be of great interest to readers of this journal. Therefore, I would recommend the publication of this article in Current Issues in Molecular Biology after addressing minor revisions.
minor comments/suggestions
- Page 13, Figure 2. Please increase the resolution of Figure 2B and 2C.
- Page 15, Figure 4. For ease of comparison, significant differences should be marked with asterisks (*, **, or ***) in Figure 4C.
- Page 17, Figure 5.Please increase the resolution of Figure 5A, 5C, and 5F.
- Page 18, Figure 6.1) The thickness of the lines compared between groups in Figure 6I is not uniform.2) The format for indicating significant differences in Figure 6 is inconsistent: Figures 6A, B, C, and E use parentheses with asterisks, while Figure 6I uses straight lines with asterisks. The author should standardize the presentation of significance indicators throughout Figure 6.
- Peptides based antagonists for cancer immunotherapy is currently among the most promising approaches for anticancer treatment. It is appropriate to cite the representative review which outlines the development of anticancer peptides for cancer immunotherapy.
- Page 1, paragraph 1, line 12, “were studied in a nude mouse model by MTT”.MTT experiments are experiments at the cellular level, and this article is presented as a study in a nude mouse model. Please outline the reason.
- Recently, oncolytic peptides which possess synergistic oncolytic-immunotherapy effect represent novel candidates for anticancer treatments. Especially, multiple oncolytic peptides have entered clinical trials for cancer immunotherapy. To help readers and potential users, the representative work on the development and anticancer application of oncolytic peptides should be cited (suggest, J. Med. Chem., 2024, 67, 3885. Acta Pharmacol. Sin., 2023, 44, 201. Bioorg. Chem., 2023, 138, 106674.).
Author Response
Manuscript Number: cimb-3822032
Title: The SMIM25-COX-2 Axis modulates the Immunosuppressive tumor microenvironment and predicts Immunotherapy Response in Hepatocellular Carcinoma.
Journal: Current Issues in Molecular Biology
Dear Reviewer 1,
Thank you for your dedicated review and constructive feedback on our manuscript. Your expertise has been instrumental in strengthening the methodological rigor and data interpretation of this work. We have carefully revised the manuscript in accordance with the feedback received. Detailed responses to each observation are enumerated below:
minor comments/suggestions
Comments 1: Page 13, Figure 2. Please increase the resolution of Figure 2B and 2C.
Response 1: We thank the reviewer for this helpful suggestion. We agree that the resolution of Figures 2B and 2C needed improvement. Accordingly, we have re-generated these figures at a higher resolution to ensure better clarity. The updated Figure 2 has been replaced in the revised manuscript.
Comments 2: Page 15, Figure 4. For ease of comparison, significant differences should be marked with asterisks (*, **, or ***) in Figure 4C.
Response 2: We thank the reviewer for this valuable suggestion. We agree that marking significant differences enhances the clarity of the figure. Accordingly, we have now added significance asterisks (*, **, ***) to Figure 4C as requested. The revised Figure 4 has been updated in the manuscript.
Comments 3: Page 17, Figure 5. Please increase the resolution of Figure 5A, 5C, and 5F.
Response 3:We thank the reviewer for this helpful suggestion. We agree that the resolution of Figure 5A, 5C, and 5F needed improvement. Accordingly, we have re-generated these figures at a higher resolution to ensure better clarity. The updated Figure 5 has been replaced in the revised manuscript.
Comments 4: Page 18, Figure 6.1) The thickness of the lines compared between groups in Figure 6I is not uniform.2) The format for indicating significant differences in Figure 6 is inconsistent: Figures 6A, B, C, and E use parentheses with asterisks, while Figure 6I uses straight lines with asterisks. The author should standardize the presentation of significance indicators throughout Figure 6.
Response 4:We thank the reviewer for their meticulous observation and valuable suggestions. We have carefully revised Figure 6 to address both points.
- We have corrected the comparison lines in Figure 6I to ensure they have a uniform thickness.
- We have standardized the format for indicating significance across all panels of Figure 6. The indicators in Figure 6I now use the same bracket-and-asterisk style as in Figures 6A, B, C, and E for consistency. The updated, fully consistent Figure 6 has been replaced in the revised manuscript.
Comments 5: Peptides based antagonists for cancer immunotherapy is currently among the most promising approaches for anticancer treatment. It is appropriate to cite the representative review which outlines the development of anticancer peptides for cancer immunotherapy.
Response 5:We sincerely thank the reviewer for this excellent suggestion. We agree that positioning our research within the promising context of peptide-based cancer immunotherapy is crucial. To address this, we have revised our Introduction to include a discussion on the development of peptide-based antagonists as a novel therapeutic strategy. We have cited several representative reviews to provide a comprehensive background for the readers.
This new content has been added to the second paragraph of the Introduction as follows: "Given the limitations of current treatments, the development of novel therapeutic modalities is of paramount importance. Among the most promising are peptide-based therapeutics, which offer unique advantages in cancer treatment due to their high specificity and favorable safety profiles. Specifically, in the field of cancer immunotherapy, a major focus is on developing peptide and small molecule modulators that target critical immune checkpoints like PD-1 and PD-L1, thereby unleashing the body’s own anti-tumor immune response ".
We deeply appreciate the reviewer’s effort in helping us improve the quality and scope of our manuscript.
Comments 6: Page 1, paragraph 1, line 12, “were studied in a nude mouse model by MTT”. MTT experiments are experiments at the cellular level, and this article is presented as a study in a nude mouse model. Please outline the reason.
Response 6:We sincerely thank the reviewer for their keen observation and for identifying this error. The reviewer is absolutely correct. Stating that the MTT assay was conducted “in a nude mouse model” was a significant writing error on our part, and we sincerely apologize for this mistake and the confusion it caused.
The MTT and wound healing assays were indeed performed in vitro to assess cellular functions, while the tumor formation study was conducted in vivo in the nude mouse model. We have now corrected this sentence in the Abstract to accurately distinguish between our in vitro and in vivo experiments.
The revised sentence in the manuscript now reads: “The biological effects of SMIM25 on cell proliferation and migration were studied in vitro using MTT and wound healing assays, while its impact on tumor growth was evaluated in vivo in a nude mouse model.”
We trust this revision now accurately reflects our study’s design.
Comments 7: Recently, oncolytic peptides which possess synergistic oncolytic-immunotherapy effect represent novel candidates for anticancer treatments. Especially, multiple oncolytic peptides have entered clinical trials for cancer immunotherapy. To help readers and potential users, the representative work on the development and anticancer application of oncolytic peptides should be cited (suggest, J. Med. Chem., 2024, 67, 3885. Acta Pharmacol. Sin., 2023, 44, 201. Bioorg. Chem., 2023, 138, 106674.).
Response 7:We are very grateful to the reviewer for this insightful comment and for providing these specific, highly relevant, and timely references. The development of oncolytic peptides is indeed a very exciting frontier in cancer immunotherapy.
Following this valuable guidance, we have incorporated a discussion on oncolytic peptides and their synergistic effects within our Introduction. We have cited the three representative articles suggested by the reviewer. This helps to underscore the urgent need for understanding immunosuppressive mechanisms (like the one driven by SMIM25) that might impact the efficacy of such novel agents.
This revision can be found in the newly added third paragraph of the Introduction as follows: "Notably, a subclass known as oncolytic peptides has shown exceptional potential, possessing a powerful synergistic effect by directly lysing cancer cells while simultaneously stimulating an anti-tumor immune response. The success of these emerging peptide-based strategies, much like existing immunotherapies, relies heavily on identifying novel molecular targets within the TME".
We deeply appreciate the reviewer’s effort in helping us improve the quality and scope of our manuscript.
We sincerely hope these revisions demonstrate the scholarly rigor required by Current Issues in Molecular Biology. The reviewer's expertise has been invaluable in enhancing the methodological transparency and data interpretability of this work. Should there be any additional requirements to facilitate the editorial process, we remain fully committed to providing prompt assistance.
Reviewer 2 Report
Comments and Suggestions for Authors
Suggestions and Comments:
1.) Only 1 cell line human hepatoma LM3 was utilized in this study. Few cell lines should be utilized to make a significant conclusion that upregulation of SMIM25 expression can be used as potential biomarker for the prognosis of HCC.
2.) The effect of COX-2 was performed on the LM3 cell line. Please explain the reason why this was not performed on the mouse model inoculated with LM-3 cells.
3.) Was a follow-up investigation performed on the patients (where 58 tissue samples were collected) to associate the HCC prognosis with the level of SMIM25 overexpression? Please explain.
Author Response
Manuscript Number: cimb-3822032
Title: The SMIM25-COX-2 Axis modulates the Immunosuppressive tumor microenvironment and predicts Immunotherapy Response in Hepatocellular Carcinoma.
Journal: Current Issues in Molecular Biology
Dear Reviewer 2,
Thank you for your dedicated review and constructive feedback on our manuscript. Your expertise has been instrumental in strengthening the methodological rigor and data interpretation of this work. We have carefully revised the manuscript in accordance with the feedback received. Detailed responses to each observation are enumerated below:
Suggestions and Comments:
Comments 1: Only 1 cell line human hepatoma LM3 was utilized in this study. Few cell lines should be utilized to make a significant conclusion that upregulation of SMIM25 expression can be used as potential biomarker for the prognosis of HCC.
Response 1: We are deeply grateful for your highly valuable feedback. We fully comprehend that relying solely on the LM3 cell line overexpressing SMIM25 in our in vivo experiments has restricted the generalizability of our research findings. Your insightful suggestion is sincerely appreciated as it points out an important aspect for us to improve.
We readily admit that using only the single LM3 cell line may not offer comprehensive representativeness. However, our choice of the LM3 cell line was based on well - considered reasons. The genetic background (HBV) and biological behavior (high metastasis) of the LM3 cell line closely align with those of our clinical patients. In our region, Guangxi, China, HCC has a distinct epidemiological profile. The majority of HCC patients here are associated with HBV infection and exhibit high metastatic characteristics, with the incidence and mortality rates ranking at the top in China. Our clinical collaborator, Dr. Bangde Xiang ’s team, has identified in clinical practice that COX - 2 might be a potential therapeutic target for HCC in Guangxi. In earlier studies, they found that among all HCC cell lines examined, the LM3 cell line had the strongest correlation between stem cell expression and COX-2 (The data were published in Medicine (Baltimore) 2015;94(44)).
We recognize that we did not utilize a broader range of cell lines to validate the universality of SMIM25’s function, and we have discussed these limitations in our study as follows: " First, our mechanistic insights, including the regulatory link between SMIM25 and COX-2 and the functional rescue by a COX-2 inhibitor, were primarily validated in vitro using a single cell line. The therapeutic potential of targeting this axis has not yet been confirmed in an in vivo setting, which represents a crucial next step for future studies. Moreover, the precise upstream regulatory mechanisms governing SMIM25 expression are still not completely understood. These factors limit the generalizability of our findings to other HCC subtypes and underscore the need for further research.".
If, in the future, we can establish primary cell lines from local HCC patients in Guangxi, it will greatly facilitate our efforts to elucidate the molecular mechanisms relevant to local HCC patients in this region.
Once again, we thank you for your significant suggestion, which has provided us with a clear direction for further improvement.
Comments 2: The effect of COX-2 was performed on the LM3 cell line. Please explain the reason why this was not performed on the mouse model inoculated with LM-3 cells.
Response 2: We sincerely thank the reviewer for this insightful question. It highlights a critical point regarding the translation of our in vitro findings to an in vivo context.
Our primary objective for performing the COX-2 inhibitor experiment was to directly and mechanistically validate that the pro-tumorigenic effects of SMIM25 overexpression are, at least in part, mediated through the COX-2 pathway at the cellular level. The in vitro system offers a controlled environment that allows for a clear and quantifiable assessment of rescue effects on specific cellular behaviors, such as migration and invasion. This approach enabled us to establish a direct “proof-of-concept” for the functional link between SMIM25 and COX-2, minimizing the confounding variables present in a complex in vivo tumor microenvironment.
We completely agree with the reviewer that an in vivo rescue experiment is the crucial next step to confirm the therapeutic potential of targeting this axis. The absence of this experiment is a significant limitation of our current study. To fully address this point, we have revised the limitation section in our Discussion to explicitly include the lack of in vivo validation.
The revised paragraph in the Discussion now reads:
"First, our mechanistic insights, including the regulatory link between SMIM25 and COX-2 and the functional rescue by a COX-2 inhibitor, were primarily validated in vitro using a single cell line. The therapeutic potential of targeting this axis has not yet been confirmed in an in vivo setting, which represents a crucial next step for future studies. Moreover, the precise upstream regulatory mechanisms governing SMIM25 expression are still not completely understood. These factors limit the generalizability of our findings to other HCC subtypes and underscore the need for further research."
This revision now clearly acknowledges both the single-cell-line limitation and the absence of in vivo therapeutic validation. We have a clear plan to investigate the efficacy of COX-2 inhibitors in our SMIM25-overexpressing mouse model in our follow-up studies.
Once again, we thank you for your valuable feedback, which has helped us to more accurately present the scope and limitations of our work.
Comments 3: Was a follow-up investigation performed on the patients (where 58 tissue samples were collected) to associate the HCC prognosis with the level of SMIM25 overexpression? Please explain.
Response 3:We sincerely thank the reviewer for raising this important question, which allows us to clarify a key aspect of our study.
Yes, a follow-up investigation was indeed performed on the 58 HCC patients from whom the tissue samples were collected. We gathered comprehensive follow-up data for these patients, which included their overall survival time, recurrence-free survival time, tumor number, BCLC stage, Edmondson-Steiner grade, microvascular invasion (MVI), CK19 expression, and HBV infection status.
We then conducted a thorough analysis to associate the expression level of SMIM25 with these prognostic outcomes. The results of this analysis are presented in detail in Figure 1F, 1G, and 1H of our manuscript.
These findings, which directly link higher SMIM25 expression to poorer prognosis in HCC patients, form a central conclusion of our work.
We hope this explanation clarifies that the prognostic significance of SMIM25 was a primary focus of our investigation. We appreciate the opportunity to highlight this point.
We sincerely hope these revisions demonstrate the scholarly rigor required by Current Issues in Molecular Biology. The reviewer's expertise has been invaluable in enhancing the methodological transparency and data interpretability of this work. Should there be any additional requirements to facilitate the editorial process, we remain fully committed to providing prompt assistance.
Reviewer 3 Report
Comments and Suggestions for Authors
Suggestions and Comments:
- Please define all acronyms as they first appear in the manuscript.
- Few minor grammatical errors and word spacing that can be fixed by the editorial team.
- The results with COX-2 is an important finding in this investigation. I suggest that the title should be modified by including it in the title.

Author Response
Manuscript Number: cimb-3822032
Title: The SMIM25-COX-2 Axis modulates the Immunosuppressive tumor microenvironment and predicts Immunotherapy Response in Hepatocellular Carcinoma.
Journal: Current Issues in Molecular Biology
Dear Reviewer 3,
Thank you for your dedicated review and constructive feedback on our manuscript. Your expertise has been instrumental in strengthening the methodological rigor and data interpretation of this work. We have carefully revised the manuscript in accordance with the feedback received. Detailed responses to each observation are enumerated below:
Suggestions and Comments:
Comments 1: Please define all acronyms as they first appear in the manuscript.
Response 1: We thank the reviewer for this important reminder. We have carefully reviewed the entire manuscript and have ensured that all acronyms are clearly defined upon their first use in the text. This includes checking the Abstract, Main Text, Figure Legends, and Tables to ensure consistency and compliance with the journal’s guidelines. We believe the revised manuscript is now much clearer for the reader.”
Comments 2: Few minor grammatical errors and word spacing that can be fixed by the editorial team.
Response 2: We thank the reviewer for pointing out the minor grammatical and spacing errors. We appreciate your valuable feedback. After incorporating all the revisions suggested by the reviewers, we have had the entire manuscript professionally proofread by a language editing service to ensure its quality meets the high standards of the journal. We believe the revised manuscript is now free of such errors and ready for publication.
Comments 3: The results with COX-2 is an important finding in this investigation. I suggest that the title should be modified by including it in the title.
Response 3:We are very grateful to the reviewer for this excellent and constructive suggestion. We completely agree that highlighting the central role of COX-2 in the title will significantly improve the manuscript’s impact and better reflect our key mechanistic findings.
Following this valuable advice, we have revised the title. We would like to propose the following new title for our manuscript:
New Proposed Title:
“The SMIM25-COX-2 Axis Modulates the Immunosuppressive Tumor Microenvironment and Predicts Immunotherapy Response in Hepatocellular Carcinoma”
We believe this revised title is more informative and powerful, as the term “Smim25-COX-2 Axis”concisely encapsulates the novel pathway we have uncovered. This change better highlights the mechanistic depth of our study, as the reviewer rightly pointed out.
We have updated the title in the revised manuscript accordingly. Once again, we thank the reviewer for helping us to improve our paper.
We sincerely hope these revisions demonstrate the scholarly rigor required by Current Issues in Molecular Biology. The reviewer's expertise has been invaluable in enhancing the methodological transparency and data interpretability of this work. Should there be any additional requirements to facilitate the editorial process, we remain fully committed to providing prompt assistance.